



# Anthropogenic and volcanic point source SO₂ emissions derived from TROPOMI onboard Sentinel 5 Precursor: first results

Vitali Fioletov[1], Chris A. McLinden[1], Debora Griffin[1], Nicolas Theys[2], Diego. G. Loyola[3], Pascal
Hedelt[3], Nickolay A. Krotkov[4], Can Li[4,5]

[1] Air Quality Research Division, Environment and Climate Change Canada, Toronto, Canada.
[2] Royal Belgian Institute for Space Aeronomy (BIRA-IASB), Brussels, Belgium
[3] Deutsches Zentrum für Luft- und Raumfahrt (DLR), Wessling, Germany
[4] Atmospheric Chemistry and Dynamics Laboratory, NASA Goddard Space Flight Center, Greenbelt, MD, USA.
[5] Earth System Science Interdisciplinary Center, University of Maryland College Park, MD, USA

*Correspondence to*: Vitali Fioletov (Vitali.Fioletov@outlook.com or Vitali.Fioletov@canada.ca)

**Abstract**.

The paper introduces the first TROPOMI-based sulfur dioxide (SO₂) emissions estimates for point sources. A total of about 500 continuously emitting point sources releasing from about 10 kT y⁻¹ to more than 2000 kT y⁻¹ of SO₂ per year, previously identified from Ozone Monitoring Instrument (OMI) observations, were analysed using TROPOMI measurements for one full year, from April 2018 to March 2019. The annual emissions from these sources were estimated and compared to similar estimates from OMI and Ozone Mapping Profiling Suite (OMPS) measurements. Note that emissions from many of these 500 sources declined significantly since 2005 making their quantification more challenging. We were able to identify 278 sources where annual emissions are significant and can be reliably estimated from TROPOMI. The standard deviations of TROPOMI vertical column density data, about 1 Dobson Unit (DU, where 1 DU = $2.69 \times 10^{16}$ molecules/cm²) over tropics and 1.5 DU over high latitudes, are larger than those of OMI (0.6-1DU) and OMPS (0.3-0.4 DU). Due to its very high spatial resolution, TROPOMI produces 12-20 times more observations over a certain area than OMI and 96 times more than OMPS. Despite higher uncertainties of individual TROPOMI observations, TROPOMI data averaged over a large area have roughly two-three times lower uncertainties compared to OMI and OMPS data. Similarly, TROPOMI annual emissions can be estimated with uncertainties that are 1.5-2 times lower than the uncertainties of annual emissions estimates from OMI. While there are area biases in TROPOMI data over some regions that have to be removed for emission calculations, the absolute magnitude of these are modest, typically within ±0.25 DU, it can be comparable to SO₂ values over large sources.



# 1 Introduction

Sulfur dioxide (SO₂) is a major air pollutant that contributes to acid rain and aerosol formation, adversely affect the environment and human health, and impacts climate. Current and accurate information about SO₂ emissions is therefore required in modern air quality and climate models (e.g., Liu et al., 2018). The majority of SO₂ emissions are related to

anthropogenic processes (e.g., combustion of sulfur-containing fuels, oil refining processes, and metal ore smelting operations), although natural processes such as volcanic eruptions and degassing also play an important role. Information about emissions from SO₂ sources is not always available or up to date, and a sizable fraction of emission sources is even missing from conventional emission inventories (McLinden et al., 2016) with satellite measurements only now being used to fill this gap. Liu et al., (2018) demonstrated that merging such satellite-based emissions estimates with a conventional bottom-up

inventory improves the agreement between the model and surface observations.

In the early 1980s, satellite measurements of backscattered radiation by the Total Ozone Mapping Spectrometer (TOMS) provided the first global estimates of SO₂ from large volcanic eruptions (Krueger, 1983). The TOMS instrument was capable of measuring backscattered solar ultraviolet (BUV) radiance at just several wavelengths. A hyperspectral instrument from the next generation (a UV-Visible imaging spectrometer), Global Ozone Monitoring Experiment (GOME) on the Earth

Research Satellite 2 (ERS-2), launched in 1995, was able to detect major anthropogenic sources (Eisinger and Burrows, 1998; Khokhar et al., 2008). The launch of the Ozone Monitoring Instrument (OMI) aboard NASA's Earth Observing System Aura satellite (Levelt et al., 2017, 2006) with high spatial resolution, of up to 13 by 24 km² at nadir, but lower at the swath edges (de Graaf et al., 2016) started a new era in satellite air quality monitoring. Data from OMI as well as from Scanning Imaging Absorption Spectrometer for Atmospheric Chartography (SCIAMACHY), on the ENVISAT, the Global Ozone Monitoring

Experiment-2 (GOME-2) on MetOp-A, -B, (Callies et al., 2000), and the Ozone Mapping and Profiler Suite (OMPS) aboard the NASA– NOAA Suomi National Polar-orbiting Partnership (Suomi NPP) were used to track SO₂ changes on the global and regional scales and estimate area- and point-source emissions (Carn et al., 2004, 2007; Fioletov et al., 2013; de Foy et al., 2009; Koukouli et al., 2016a, 2016b; Krotkov et al., 2016; Lee et al., 2009; Li et al., 2017; McLinden et al., 2012, 2014; Rix et al., 2012; Nowlan et al., 2011; Thomas et al., 2005; Zhang et al., 2017). Moreover, OMI measurements were used to evaluate

the efficacy of cleantech solutions in reducing SO₂ emissions from industrial sources (Fioletov et al., 2013, 2016; Ialongo et al., 2018; Song and Yang, 2014).

There are two major types of UV-visible SO₂ retrieval algorithms for nadir viewing instruments. The traditional Differential Optical Absorption Spectroscopy (DOAS) scheme is based on the approach where absorption cross sections of relevant atmospheric gases are adjusted by a nonlinear least squares fit procedure to the log ratio of a measured earthshine

spectrum and a reference spectrum in a given wavelength interval (Theys et al., 2015). The DOAS algorithm requires information about the absorption spectra of all trace gases, non-elastic Rotational Raman scattering (Ring effect) and information about the instrument characteristics. The uncertainties of the DOAS algorithm arise from inaccurate modeling of the various physical processes in solar light absorption and scattering (e.g., Ring effect, surface properties) as well as artifacts



in the radiance measurements (e.g., stray light, wavelength shift). An alternative approach is used in the principal component analysis (PCA) algorithm. Instead of attempting to model all various factors other than $SO_2$, the PCA algorithm replaces them with characteristic features derived directly from the measurements over locations where no $SO_2$ is expected (Li et al., 2013, 2017, 2019). When applied to OMI measurements, both DOAS and PCA algorithms produce similar results, however the PCA

algorithm-based data show reduced data scattering and smaller biases comparerd to the DOAS algorithm-based data (Fioletov et al., 2016).

The launch of TROPOspheric Monitoring Instrument (TROPOMI) on board Copernicus Sentinel-5 Precursor in October 2017 made it possible to monitor atmospheric pollutants with an unprecedented spatial resolution, 3.5 by 7 km$^2$ (Veefkind et al., 2012), that is at least 12 times better that resolution of OMI. Since August 6$^{th}$ 2019 the spatial resolution was

further reduced in the flight direction, the TROPOMI ground pixel size is now 3.5 by 5.5 km$^2$. It has already been demonstrated that TROPOMI can successfully monitor trace gases such as ozone (Garane et al., 2019), $NO_2$ (Griffin et al., 2019), HCHO (de Smedt et al., 2018), CO (Borsdorff et al., 2019), $CH_4$ (Hu et al., 2018), and even BrO (Seo et al., 2019) as well as cloud properties (Loyola et al., 2018). The operational TROPOMI $SO_2$ retrieval algorithm utilizes the DOAS approach (Theys et al., 2017) and early observations demonstrated the benefits of high spatial resolution for monitoring volcanic plumes (Hedelt et

al., 2019; Theys et al., 2019; Queißer et al., 2019). However, these first studies were focused on relatively high volcanic $SO_2$ levels. In this study, we perform analysis of TROPOMI $SO_2$ observations that include smaller anthropogenic and volcanic degassing sources. We applied a previously developed technique (Fioletov et al., 2015) to estimate $SO_2$ emissions from TROPOMI observations. About 500 $SO_2$ sources, previously identified using OMI 2005-2015 data (Fioletov et al., 2016), were examined and their emissions were estimated using TROPOMI data and then compared to emission estimates from OMI

and OMPS.

## 2 Data Sets

### 2.1 Satellite $SO_2$ VCD data

The TROPOMI instrument onboard of the Sentinel-5 Precursor (S5P) satellite was launched on October 13, 2017. TROPOMI has the smallest spatial footprint of 3.5 by 7 km$^2$ (3.5 by 5.5 km² after August 2019) among the instruments of its class

(Veefkind et al., 2012). TROPOMI measures spectra of backscattered solar light at 450 cross-track positions (or pixels) and provides daily global coverage. TROPOMI $SO_2$ Level 2 (/PRODUCT/sulfurdioxide_total_vertical_column) data processed with the S5P operational processing system UPAS version 01.01.05 (Theys et al., 2017) were used in this study. In the first step of the algorithm, $SO_2$ slant column densities (SCDs), representing the effective optical-path integral of $SO_2$ concentration, were retrieved using the DOAS method. An additional background correction was applied to remove possible biases in SCDs

after the spectral retrieval step. The spectral fitting was done using the 312–326 nm window, although two other spectral windows (325–335 nm and 360–390 nm) were used for retrievals in cases of very high volcanic $SO_2$. The final product, the $SO_2$ vertical column densities (VCDs) were calculated from SCDs using conversion factors (air mass factors). VCDs represent





the number of $SO_2$ molecules (or total mass) in an atmospheric column per unit area. VCDs are commonly reported in Dobson Units (DU) where 1 DU = $2.69 \times 10^{16}$ molecules/cm². The standard TROPOMI $SO_2$ data product additionally includes VCDs calculated for three volcanic scenarios: when a 1 km thick plume is located at ground level, at 7 km and 15 km. In this study we are focussed on anthropogenic and degassing volcanic emissions and used only data corresponding to the ground level

plume.

OMI, a Dutch-Finnish UV-Visible wide field of view nadir-viewing spectrometer onboard NASA's Aura satellite was launched on July 15, 2004 (Schoeberl et al., 2006). Originally, it was able to provide daily global coverage with the resolution up to 13 by 24 km² at nadir (de Graaf et al., 2016; Levelt et al., 2006), but now about a half of its pixels are affected by a field-of–view blockage and stray light (the so-called "row anomaly") and $SO_2$ cannot be retrieved successfully from those

pixels. The OMI detector has 60 cross-track positions. In our previous studies (Fioletov et al., 2016; McLinden et al., 2016), we excluded data from the first 10 and last 10 cross-track positions from the analysis to limit the across-track pixel width from 24 km to about 40 km. However, due to row anomaly, this currently limits the number of available pixels to 15-20. We found that excluding only the first and the last 5 cross-track positions does not change the emission estimates noticeable but reduces their uncertainties, so only the first and last 5 pixels were excluded from the current analysis. NASA operational Planetary

Boundary Layer (PBL) $SO_2$ Level 2 data product was used in this study (OMSO2; Li et al., 2019a). This data product is produced with the Principal Component Analysis (PCA) algorithm (Li et al., 2013; 2017). The 310.5–340 nm spectral window was used for $SO_2$ retrievals. Detailed information on the OMI PCA $SO_2$ data sets and its characteristics is available elsewhere (Krotkov et al., 2016; McLinden et al., 2015). It should be noted that OMI DOAS algorithm-based data product is also available (Theys et al., 2015). While the results of the two algorithms are somewhat different, particularly in large-scale biases, emission

estimates from the two algorithms demonstrate very similar results (Fioletov et al., 2016).

OMPS Nadir Mapper on board the Suomi National Polar-orbiting Partnership (or Suomi NPP) satellite operated by NASA and NOAA was launched in October 2011. The standard NASA OMPS $SO_2$ data product (NMSO2-PCA-L2) is based on the same PCA algorithm as the NASA OMI data product (Li et al., 2019b; Zhang et al., 2017). OMPS has a lower spatial resolution than OMI, 50 km by 50 km, but better signal-to-noise characteristics. OMPS $SO_2$ VCD data are retrieved for 35

cross-track positions. Similar to OMI data analysis, large OMPS pixels at the edges of the swath (rows < 2 or > 33) were excluded. Both OMI and OMPS $SO_2$ data are retrieved with the same PCA algorithm, and emissions estimates for the two satellite instruments are similar, although OMPS tends to miss or underestimate emissions from small sources (Zhang et al., 2017).

Suomi NPP and S5P are on the same orbit 3.5 minutes apart and cross the Equator at about 13:30 local time. Aura is on a

similar polar orbit and cross the Equator at about 13:45 local time. Therefore, we can assume that there is no difference in the measurements of the three satellite instruments related to diurnal variations of $SO_2$. TROPOMI operational $SO_2$ data record starts in April 2018. In order to have one full year of data, we analyzed TROPOMI, OMI, and OMPS data for the period from April 2018 to March 2019.



For emission estimates, we examined SO$_2$ values within a 300 km radius from each emission source listed in the SO$_2$ point source catalogue (Fioletov et al., 2016). There are about 500 sources in the catalogue, however many sources emitting SO$_2$ in the first years of OMI operation, were below the OMI sensitivity level in 2018: either closed or now produce substantially reduced emissions due to scrubber installation. The most recent version of the SO$_2$ emission catalogue is available

from NASA public archive (Fioletov et al., 2019) and at https://so2.gsfc.nasa.gov/measures.html.

## 2.2 Air mass factors and data filtering

Data filtering was applied to OMI, OMPS, and TROPOMI SO$_2$ data before the analysis. The current retrieval algorithms are optimized for low (0.05) surface albedo, therefore pixels that corresponds to snow-covered high albedo surfaces were excluded from the analysis. Measurements taken at high solar zenith angles (more than 70°) were also excluded. Only clear-sky data,

defined as having a cloud radiance fraction (across each pixel) less than 20% were used. Negative SO$_2$ values that are less than -3 DU were also excluded. To eliminate cases of transient volcanic SO$_2$, periods when high SO$_2$ values were caused by eruptions, days then the highest 10% of SO$_2$ values were above a limit near the analysed site were excluded from the analysis. The limit depends on the emission strength and varies from 6 DU for sources emitting less than 100 kt per year to 15 DU for sources emitting >1000 kt per year.

Information on airmass factors (AMFs) is required to convert TROPOMI SCDs to VCDs. AMFs depend on SO$_2$ vertical profile shape, solar zenith angle, observation geometry, total ozone absorption, clouds, and surface reflectivity. In the operational TROPOMI dataset, TM5 model calculations were used to obtain a-priori SO$_2$ vertical profile to calculate AMF for each TROPOMI pixel. The model estimates rely on "bottom-up" emission inventories derived from economic activity data and SO$_2$ emission factors for known sources, so that in the case of a missing source in the inventory, the model SO$_2$ profile

shape would be representative of clean background areas, causing calculated AMFs to be biased high and VCDs being underestimated over that source.

The PCA algorithm uses spectrally dependent SO$_2$ Jacobians instead of AMFs. To make it consistent to the previous operational OMI Band Residual Difference (BRD) algorithm, the present PCA algorithm assumes the same fixed conditions that correspond to a typical summertime conditions in the eastern USA and PCA retrievals can therefore be interpreted as

having an effective AMF of 0.36 as in the BRD algorithm (Krotkov et al., 2006). However, a constant AMF does not represent conditions such as high elevations or enhanced aerosol loading. As in our previous studies (Fioletov et al., 2016; McLinden et al., 2016) a single site-specific AMF was calculated for each source (McLinden et al., 2014) and applied to both OMI/OMPS and TROPOMI estimated emissions.

As one of the main goals of this study is to compare TROPOMI SO$_2$ data and emissions estimates to those from OMI

and OMPS, we used a constant AMF of 0.36 for illustration maps while for the emissions estimates, we converted TROPOMI SO$_2$ SCDs to VCDs using the same site-specific AMFs thereby removing it as a potential source of variability. It should be also noted that the spectral fitting window used in the TROPOMI algorithm is different from the window in the PCA algorithm. However, we estimated that that effect is small (under 10%) compared to other sources of uncertainties.



The SO$_2$ absorption cross section has a moderate temperature dependence, with absorption increasing for higher temperatures, and there is a difference in how this dependence was handled in TROPOMI and OMI/OMPS retrievals. In the TROPOMI spectral fit, an SO$_2$ cross section at 203°K was used and then the retrieved VCDs were adjusted by applying an AMF correction factor using temperature from the European Centre for Medium-Range Weather Forecast (ECMWF) operational model (Theys et al., 2017). The OMI/OMPS retrieval algorithm uses the SO$_2$ cross section at 293°K (Krotkov et al., 2006) without any adjustment. In this work, for consistency, we converted TROPOMI SO$_2$ SCDs to VCDs using the same AMF approach as we utilized for OMI and OMPS (without any temperature adjustment), meaning the obtained VCDs now correspond to 203 K. To remove this systematic difference with OMI/OMPS data, we increased the TROPOMI SO$_2$ VCDs by 22% (see Theys et al., (2017), their Figure 6, for justification).

## 2.5 Wind and snow data

The emission estimation algorithm requires wind data. As in several previous studies (Fioletov et al., 2015; McLinden et al., 2016), European Centre for Medium-Range Weather Forecasts (ECMWF) reanalysis data (Dee et al., 2011) (http://apps.ecmwf.int/datasets/) were extracted for every satellite pixel. Wind profiles are available every 6 hours on a 0.75° horizontal grid and are interpolated in time and space to the location of each satellite pixel center. U- and V- (west-east and south-north, respectively) wind-speed components were averaged for 1-km thick layers and the winds for the layer that corresponds to the site altitude was used. The Interactive Multisensor Snow and Ice (IMS) Mapping System data (Helfrich et al., 2007) were used to screen out pixels over snow-covered surface with high albedo.

## 3 TROPOMI SO$_2$

For brevity, from this point we refer to "SO$_2$ VCD" as simply to "SO$_2$". It can be expected that a smaller pixel size of TROPOMI would yield a lower signal to noise level. Figure 1 shows the standard deviation of SO$_2$ values at four sites, each located at different latitudes, as a function of the TROPOMI cross-track position. The selected sites have relatively small SO$_2$ emissions, so the standard deviations are determined by the instrumental noise and possible retrieval uncertainties. The standard deviations at the 20 cross-track positions at the edges of the swath are particularly high due to shorter exposure time, which motivated our decision (in addition to a larger footprint) to exclude them from the analysis. There is also a clear increase in the noise from low to high latitudes with the noise standard deviations at a sub-polar site nearly double compared to tropical sites. Outside the tropical belt, there is also some seasonality in the standard deviation values with higher values occurring in winter and lower in summer (not shown).

The standard deviation of SO$_2$ retrievals for the three satellite instruments as a function of latitude is shown in Figure 2 for the period from April 2018 to March 2019. The plot is based on satellite measurements over clean areas (150 km-300 km distance from the catalogue source locations) and represent background noise levels of SO$_2$. Large sources with annual SO$_2$ emissions above 1000 kt per year where the high standard deviations are likely to be influenced by the SO$_2$ variability itself





were excluded from this analysis. Sources inside the South Atlantic Anomaly (SAA) region were also excluded. The standard deviations of TROPOMI data (about 1 DU over tropics and 1.5 DU over high latitudes) are larger than those of OMI (0.6-1DU) and OMPS (0.3-0.4 DU) data. The standard deviations are particularly large (1.6-2.2 DU) for the first and the last 20 pixels in the TROPOMI 450 pixels-wide swath, which were excluded from further analysis.

As Figure 2 shows, the standard deviations ($\sigma$) for TROPOMI are roughly 1.5 time larger than OMI, and 3 times larger than OMPS. However, the pixel size for TROPOMI is much smaller, and so the number of observations ($n$) over the same area for TROPOMI is 12 and 96 times that of OMI and OMPS, respectively. Considering these two factors, and assuming the standard error is proportional to $\sigma/\sqrt{n}$ (assuming that the errors of individual pixels are not correlated), then the uncertainty of a TROPOMI average will be roughly a factor of 2 smaller than OMI and a factor of 3 smaller than OMPS In fact, due to

the OMI row anomaly, the number of TROPOMI pixels over the same area is now a factor of 20 larger.

The global distribution of mean $SO_2$ from TROPOMI (smoothed using oversampling technique or pixel averaging technique with a 30 km radius, see e.g., Fioletov et al., (2011), Sun et al., (2018)) is very similar to that from OMI and OMPS (Figure 3). All three instruments clearly show elevated values over the Persian Gulf, China, Mexico, and India, as well as many anthropogenic "hotspots" such as Norilsk (Bauduin et al., 2014; Khokhar et al., 2008), a cluster of power plants in South

Africa, and large volcanic sources such as Kilauea, Hawaii, and Ambrym, Vanuatu. All three satellite data sets shown in Figure 3 do not demonstrate large biases seen in the older versions of OMI, GOME-2, and SCIAMACHY data (see Fioletov et al., (2013), their Figure 1). Except for the hotspot-affected areas, $SO_2$ values from all three instruments are typically within the ±0.25 DU range. It is also interesting to note that the South Atlantic Anomaly (SAA), an area of increased flux of energetic "solar wind" particles that may intercept instruments in low-Earth orbits such as these, significantly increases the uncertainties

of OMI and OMPS data (as well as data from GOME-2 and SCIAMACHY), but has little effect on TROPOMI data.

There are, however, still some differences in the absolute values between OMI, OMPS, and TROPOMI over some regions. Zoomed-in plots of mean $SO_2$ over four regions of elevated $SO_2$ values, Northern China, India, Mexico, and Iran are shown in Figure 4. TROPOMI $SO_2$ means are, in general, higher than OMI and OMPS values over these regions suggesting possible biases in TROPOMI data. The spatial scale of these biases (thousands of km) is larger than the scale of elevated $SO_2$

values from a typical industrial source (one-two hundreds of km), so we will call them "large-scale biases". Note that the biases are very small, only 0.1-0.2 DU, however even such small biases could affect emission estimates since the $SO_2$ enhancements from many sources are really tiny, a few tenths of a DU. These large-scale biases are common in satellite $SO_2$ retrievals. Their magnitude often depends on the retrieval algorithm and the same satellite measurements (i.e., calibrated level 1B data) processed with different $SO_2$ algorithms produce different biases. For example, GOME-2 data processed with the original

operational algorithm (Valks and Loyola, 2008) had larger biases than the $SO_2$ data product based on a direct fitting method developed by the Harvard-Smithsonian Center for Astrophysics, Cambridge, Massachusetts, (Nowlan et al., 2011): see Figure 1 in Fioletov et al., (2013). The origin of such biases is not always known although an imperfect removal of the very strong ozone absorption, that itself depends on stratospheric temperature and shape of the ozone profile, could be one of the contributing factors.



OMI data processed with a DOAS algorithm (Theys et al., 2015) that is similar to the present TROPOMI algorithm, also had larger biases over some areas than seen in the PCA-based data (Fioletov et al., 2016). However, as was also noted by Fioletov et al., (2016), both algorithms produce very similar results if the large-scale biases are removed, for example, by comparing up-wind and down-wind values around an $SO_2$ emission source. In case of large-scale biases in the area with

multiple sources, the bias can be accounted for by introducing functions that change slowly with latitude and longitude as suggested by Fioletov et al., (2017). This multi-source algorithm accounts for the bias using Legendre polynomials of latitude and longitude and their products and the emissions using functions that represent plumes from individual sources. As an example, Figure 5 shows original data from TROPOMI, OMI and OMPS over Europe and the same data with the local bias removed using 6[th] degree polynomials (see Fioletov et al., (2017) for details). As Figure 5 suggests, large-scale biases seen in

the original TROPOMI data are removed by this statistical fitting procedure. Note that OMPS data also show some large-scale biases over that region. The maps with the large-scale bias removed look very similar for all three satellite data sources and all the major $SO_2$ hotspots are clearly seen. Note that there is practically no bias in OMI data over southern Europe hence OMI data with and without bias removed appear very similar.

The problem of the bias in TROPOMI data as well as in data from other satellites requires further investigation and

probably improvements of the $SO_2$ algorithms. In case of TROPOMI, we saw such biases over many major areas of interest: China, India, Europe, and the Persian Gulf. The biases are often larger than the signals from emission sources that creates an impression that TROPOMI values over such sources are larger than those from OMI. It also appears that the biases are larger in winter-fall than in summer and are also larger over water. It will be possible to investigate time dependence of these biases as more TROPOMI data become available.

As mentioned, the uncertainties of TROPOMI data averaged over a certain area are 2-3 times smaller than those for OMI and OMPS. Due to its very high spatial resolution, a single year of TROPOMI can provide as much information on the $SO_2$ distribution around hotspots as compared with several years of OMI or OMPS. Figure 6 (top) shows the mean $SO_2$ over Bosnia & Herzegovina and Serbia in 2018 from OMI and TROPOMI and over the 2014-2018 period from OMI, using oversampling technique (see Sun et al. (2018) and references therein). In these countries, $SO_2$ emissions were not under the

same strict emission cutting regulations as in the EU countries. Emissions from the power plants shown in Figure 6 remained nearly constant in 2014-2018. A simple version of the oversampling technique was applied where a geographical grid is established around the source and the mean value of all satellite pixels centred within a 30 km radius from each grid point is calculated. As the mean is calculated, the standard error of the mean can be also calculated and used to evaluate the significance of that mean value by analysing the ratio of the mean value to its standard error. Figure 6 shows both, the mean values (the top

row) and the ratios (the bottom row). Although individual TROPOMI $SO_2$ values are noisier than for OMI, the much larger volume of TROPOMI data contributing to the mean makes it appear less noisy than a one-year OMI map and only a five-year OMI average demonstrates a TROPOMI-like level of noise. This is further confirmed by the ratio maps (Figure 6, bottom): TROPOMI one-year ratios are as high as 25, while OMI one-year ratios are under 10 and only five-year ratios are close to those for one-year TROPOMI values.





Although averaging of multiple years of OMI data can produce the same or even higher signal to noise ratio as one year of TROPOMI data, OMI cannot provide the same level of detail as TROPOMI due to the difference in the instrument spatial resolutions. The high spatial resolution of TROPOMI also makes it possible to see individual sources in areas where multiple sources are in a close proximity. As an example, Figure 7 shows the mean $SO_2$ over a cluster of power plants in South

Africa using one year of TROPOMI data and the entire (2005-2019) available record of OMI. For this plot, the pixel averaging with a 10 km radius was used (smaller radiuses make the OMI map too noisy to see individual sources). Although we used a very small radius for averaging, it is hard to distinguish individual sources in the OMI map, while on the one-year TROPOMI map they appear as local maxima or "hotspots".

High TROPOMI spatial resolution makes it possible in some cases to resolve an individual, persistent $SO_2$ plume. As

an example, the mean $SO_2$ over Hawaii for the period from April 2018 to March 2019 is shown in Figure 8. The source, Kilauea volcano, is located at 1200 m above sea level, while the mountains north and north-west of the volcano are as high as 4000 m. The area is dominated by easterly winds. TROPOMI data demonstrate that, on average, elevated $SO_2$ values are not observed above the volcano peak. This means that the symmetrical modified Gaussian plume model used for emissions calculations may not describe the actual plume very well in this particular case. OMI data with their lower spatial resolution do not really

show these features of the $SO_2$ distribution.

## 4 Emissions estimates

A method developed to estimate emissions from point sources from OMI data (Fioletov et al., 2015) was applied here to TROPOMI, OMI, and OMPS data. The method is based on a fit of satellite data to an empirical plume model developed to describe the $SO_2$ spatial distribution near emission point sources. First, satellite measurements are merged with wind data and

the rotation technique applied (Pommier et al., 2013; Valin et al., 2013), so the satellite data can be analysed assuming that the wind always has the same direction. Then, emissions and lifetimes were estimated using the exponentially modified Gaussian fit (Beirle et al., 2014; Fioletov et al., 2015; de Foy et al., 2015) appropriate for a near-point source. The fitted plume model depends on three parameters, total mass ($\alpha$) near the source, the lifetime or, more accurately, decay time ($\tau$), and the plume width ($\sigma$). Finally, the emission strength ($E$) is calculated from $\tau$ by $E=\alpha/\tau$. For each source, all three parameters can be derived

from a fit using a nonlinear regression model, but when doing so the uncertainties in the non-linear parameters ($\tau$ and $\sigma$) are often large. To minimize this uncertainty, all emissions were derived using a mean $\tau$ and $\sigma$ determined by averaging over values obtained from the non-linear fits. Thus, only one parameter ($\alpha$) is derived from the fit that turns the algorithm into a simple linear regression model (Fioletov et al., 2016).

The three parameters for each of the three satellite instruments were estimated using April 2018 – March 2019 data.

It can be expected that that the lifetime $\tau$ that characterizes the plume decay is the same for all three instruments. Indeed, we found that the average value of $\tau$ is about 6 hours for all three of them. The plume width $\sigma$ depends on the instrument pixel size and is expected to be different. We estimated that, as in the previous study (Fioletov et al., 2016), $\sigma$ is about 20 km for



OMI. For OMPS with its larger pixels, the average $\sigma$ value is about 25 km. For TROPOMI, the average value of $\sigma$ is about 15 km. However, many SO$_2$ sources are not really "point sources". Industrial sources are often comprised of several individual facilities located a few km apart. For example, in Norilsk, there are three major smelting factories located 8-10 km apart. For relatively large OMI pixels, this typically does not affect $\sigma$ calculations. For TROPOMI, however, we can see that for real

"point" sources $\sigma$ is smaller, about 10 km, than for sources with multiple facilities. Our sensitivity study suggests that a change of sigma from 15 km to 10 km reduces the emission estimates by about 20%. A better characterization of the emission sources will be required in the future in order to improve emission estimates for sources with multiple facilities.

The calculations were performed in the same manner as the original study for OMI data (Fioletov et al., 2016). To avoid impacts of local biases on emission estimates, the mean SO$_2$ upwind from the source was subtracted from the data. To

minimize interference from other sources, the fitting was done for a rectangular area that for small sources (<100 kt SO$_2$ yr$^{-1}$) spreads ±30 km across the wind direction, 30 km in the upwind direction and 90 km in the downwind direction. For large sources (>1000 kt SO$_2$ yr$^{-1}$), these numbers are 90 km, 90 km and 270 km respectively. Only pixels with associated wind speeds between 0.5 and 45 km h$^{-1}$ were used for the fitting. The overall uncertainty of the method is about 50%. There are several factors that contribute to the emission estimate uncertainty, however the major contributors, uncertainties in AFMs and

$\tau$, appear as scaling factors that affect TROPOMI, OMI, and OMPS-based estimates the same way. Additional information about the algorithm and uncertainly analysis can be found in (Fioletov et al., 2016).

We examined all sources listed in the catalogue (Fioletov et al., 2016) and calculated emissions for the period from April 2018 to March 2019 and their uncertainties using data for the three satellite instruments. It should be mentioned that although the catalogue contains about 500 sources, many were either closed or their emissions declined significantly due to

several possible factors such as the installation of scrubbers and reduction in coal consumption. This includes most of the sources in the USA, European Union and many sources in China. Volcanic degassing emissions also vary with time (Carn et al., 2017) and some of the volcanoes that were active at the beginning of OMI operations did not emit high amounts of SO$_2$ in 2018-2019. Therefore, a decline in the number of catalogue sources detectable by TROPOMI is not entirely unexpected. The map of catalogue sources that are detectable from one year of TROPOMI data is shown in Figure 9. The following criteria

were used to identify a source as detectable: (a) the sources should have an emission to uncertainty ratio exceeding 5 or (b) being between 4 and 5, but with a clear "hotspot" at the source with a downwind tail. There are only 20 sites in the (b) category, and we examined them on the case-by-case basis. A total of 278 sites including 150 Power Plants, 19 Smelters, 41 Oil and Gas industry-relates sources, and 68 volcanoes with annual emissions from 10 to 2000 kt SO$_2$ yr$^{-1}$ that satisfy these conditions were detected.

Scatter plots of TROPOMI, OMI, and OMPS-based emission estimates for all SO$_2$ catalogue sites are shown in Figure 10 a and b. Emissions estimates from OMI are on the horizontal axis of the both panels. Both OMPS and TROPOMI emissions estimates show a good agreement with OMI estimates for sources with estimated emissions above 50-60 kt SO$_2$ yr$^{-1}$ (calculated as an average of emission estimates from the three instruments). For them, the correlation coefficients are about 0.97 for both





instruments. However, the correlation coefficient is only 0.3 if calculated just for sources that emit less than 60 kt $SO_2$ $yr^{-1}$. There is practically no systematic biases between estimates from the three instruments.

Not surprisingly, statistical uncertainties of the emissions estimates from the three satellite instruments are also highly correlated (Figure 10 c and d). In general, the OMPS-based emission uncertainties are slightly larger than those based on OMI data. The OMI-based emission uncertainties are almost always larger than those from TROPOMI data.

The relative TROPOMI emission uncertainties are lower than those from OMI. To illustrate that, Figure 10 e and f show scatter plots of the ratios of their signal to uncertainty ratios. For very large sources ($>1000$ kt $yr^{-1}$), the emission to uncertainty ratio is dominated by $SO_2$ variability, not by the noise in satellite data. For example, $SO_2$ emissions from volcanic sources could be very different from day to day. Even if the emissions are fairly constant, different weather conditions (e.g., dry conditions vs. rain) affect the $SO_2$ dispersion patterns observed by satellites. For these very large sources, such $SO_2$ variability is larger than instrumental errors and the emission to uncertainty ratio is nearly the same for all three instruments. For smaller sources, however, measurement uncertainties play a bigger role. For OMPS, the ratios are mostly below the 1:1 line meaning that the uncertainties of OMPS-based emission estimates are higher than those based on OMI data. It is opposite for TROPOMI, where the ratios are mostly above the 1:1 line. Moreover, for medium-size and small sources, the ratios group around the 1.5:1 and 2:1 lines meaning that the TROPOMI emission estimate uncertainties are 1.5-2 times lower than those for OMI.

As all three satellite data sets can provide relatively independent emission estimates, the present satellite based $SO_2$ emission inventory could be further improved by combining emission estimates from the three sources. Due to its high resolution, and hence lower detection limit, TROPOMI can potentially identify many more sources than OMI and OMPS and then obtain emission estimates for them. An exhaustive analysis of this is beyond the scope of this paper. However, as an example of the sizable advantage offered by TROPOMI, Figure 11 shows the mean TROPOMI $SO_2$ distribution (from April 2018 to March 2019) at the border between Iran and Turkmenistan. The biggest source is the Khangiran gas refinery (1), an Iranian source that is included in the Catalogue. The second largest source is located near Mary, Turkmenistan (2), is related to gas exploration. The LANDSAT satellite images show that the source was build in 2012-2014. While one year of OMI data shows a signal from that region, they hardly can point to the source location. TROPOMI data clearly show a hotspot in both mean $SO_2$, as shown, and the high signal-to-noise ratio (not shown), right at the source location. Moreover, there are two other sources that can be resolved by TROPOMI. One of them, located east of Khangiran, could be related to two power plants (Toos and Ferdosi (3)) that are 1 km apart. This source can also be used as an illustration of the difference in emission uncertainties between TROPOMI and OMI/OMPS. TROPOMI-based emission estimates for this source is 14 kt $SO_2$ $yr^{-1}$ with the standard error of 2.8 kt $yr^{-1}$, five times lower than the emission strength itself. The standard errors of OMI and OMPS-based emission estimates are 6.1 kt $yr^{-1}$ and 7.1 kt $yr^{-1}$ respectively, 2-3 times that of TROPOMI.





## 5 Summary and discussion

The first analysis of TROPOMI near-surface $SO_2$ for the period from April 2018 to March 2019 reveals global distributions and features very similar to those seen from OMI and OMPS: elevated values over the Persian Gulf, India, China; major "hotspots" over Norilsk, Russia; South Africa; major volcanic persistent sources such as Kilauea, Hawaii, and Ambrym, Vanuatu. Outside the areas affected by these "hotspots" all three instruments typically demonstrate low background $SO_2$ values within ±0.25 DU.

Over clean areas the spatial standard deviations of TROPOMI data (about 1 DU over tropics and 1.5 DU over high latitudes) are larger than those of OMI (0.6-1 DU) and OMPS (0.3-0.4 DU) data. However, despite higher uncertainties of individual TROPOMI pixels, spatially averaged TROPOMI data over respective field-of-views have uncertainties that are 2-3 times smaller than that from OMI and OMPS data. As a result, annual mean $SO_2$ maps smoothed by a spatial filtering appears less noisy than corresponding OMI maps. In terms of the signal to noise ratio, TROPOMI smoothed one-year mean values have the same uncertainties as 4-5 years of smoothed mean values based on OMI data.

We tested about 500 $SO_2$ sources previously detected from OMI data in 2005-2015, however, many of these sources emitted much less $SO_2$ in 2018-2019 than in the beginning of OMI operation. That includes, for example, almost all US sources, many sources in Europe and China. We were able to identify 278 sources where annual (from April 2018 to March 2019) emissions can be estimated from TROPOMI data. Their emissions are in the range from 10 to 2000 kt $SO_2$ yr$^{-1}$.

Currently TROPOMI is able to provide point-source $SO_2$ emission estimates that have 1.5-2 times lower uncertainties than those from OMI, but it is less than expected taking into account that the number of useful TROPOMI pixels is about 20 times higher than that of OMI. If the statistical uncertainties are inversely proportional to the square root of the number of averaged pixels and their uncertainties are the same, then it is expected that the standard errors for TROPOMI emissions estimates should be about 4-5 times lower than those for OMI. Taking into account that the $SO_2$ uncertainties of individual TROPOMI pixels are 1.5-2 times larger than those for OMI one can expect that TROPOMI emission uncertainties would be 2.5-3.3 times lower than those for OMI, higher than the values of 1.5-2 that we derived directly from emission estimates. This may suggest that errors in individual TROPOMI pixels are correlated, for example, due to large-scale biases.

There are larger scale spatial biases in TROPOMI data over some areas that appear to be larger than similar biases in OMI and OMPS data. While the absolute magnitude of the biases is not very large, 0.1-0.2 DU, it can be comparable to the $SO_2$ enhancements over large sources. Due to these biases, $SO_2$ values over some sources may appear larger in TROPOMI data than in OMI and OMPS data. If, however, such biases are removed by, for example, a statistical fitting procedure, TROPOMI annual mean $SO_2$ maps are very similar to OMI and OMPS. It also appears that these TROPOMI biases have a larger amplitude in winter and fall, although it is hard to say that this is a repeatable seasonal effect based on just one year of data.

Biases are very common in early versions of all satellite $SO_2$ products and currently their origin is still not completely clear. The very small $SO_2$ absorption signal in the UV needs to be detected against a large contribution from the ozone



absorption. The latter is a strong function of stratospheric temperature (and hence ozone profile). Any imperfection in any of these parameters may yield a bias in retrieved $SO_2$. This, however, does not explain biases in the tropical region where ozone variability is low. Development of a PCA-type algorithm for TROPOMI could help reducing these biases and improve the overall quality of the data. An improved version of the TROPOMI processing algorithm that includes a PCA component may

produce a data product with smaller biases and lower noise than the present operational algorithm. Such improved algorithm is now under development. Preliminary test TROPOMI $SO_2$ retrievals applying a PCA-based algorithm have shown some promises, and work is underway to better understand algorithmic differences between DOAS and PCA.

## 6 Data availability

OMI and OMPS PCA $SO_2$ data used in this study have been publicly released as part of the Aura OMI Sulfur Dioxide Data Product (OMSO2) and can be obtained free of charge from the Goddard Earth Sciences (GES) Data and Information Services Center (Li et al., 2019 a, b). TROPOMI data are freely available from the European Union Copernicus Sentinel 5p data hub (https://s5phub.copernicus.eu; http://www.tropomi.eu).

*Competing interests.* The authors declare that they have no conflict of interest.

*Author contribution.* FV, CM, and DG worked on the emission methods and data analysis. NT, DL, and PS worked on the TROPOMI SO2 retrieval and data processing. NK and CL worked on the OMI/OMPS SO2 retrieval and data processing. The publication was prepared by VF, and all other authors contributed to the discussion of the paper.

*Acknowledgments.* We acknowledge the NASA Earth Science Division for funding OMI and OMPS $SO_2$ products development and analysis. The Dutch Finnish-built OMI instrument is part of the NASA EOS Aura satellite payload. The OMI project is managed by the Netherlands Royal Meteorological Institute (KNMI). The KNMI activities for OMI are funded by the Netherlands Space Office.This paper contains modified Copernicus Sentinel data. We acknowledge financial support

from DLR programmatic (S5P KTR 2472046) for the development of TROPOMI retrieval algorithms. The Sentinel 5 Precursor TROPOMI Level 1 and Level 2 products are processed at DLR with funding from the European Union (EU) and the European Space Agency (ESA). Nicolas Theys acknowledges financial support from ESA S5P MPC (4000117151/16/I-LG) and Belgium Prodex TRACE-S5P (PEA 4000105598) projects.



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

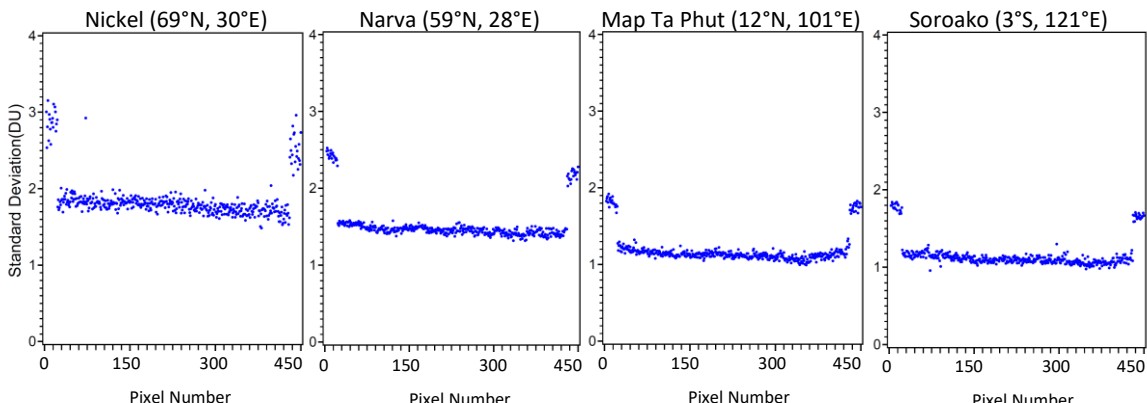

**Figure 1**. The $SO_2$ standard deviation as a function of the TROPOMI cross-track position (pixel number) at four sites illustrate a decline from high to low latitudes for the period from April 2018 to March 2019. The selected four sites represent sources with very low $SO_2$ emissions and therefore the standard deviations represent the measurement uncertainties. The $SO_2$ values retrieved at the first and last 20 pixels have noticeably higher standard deviations and are excluded from the analysis.





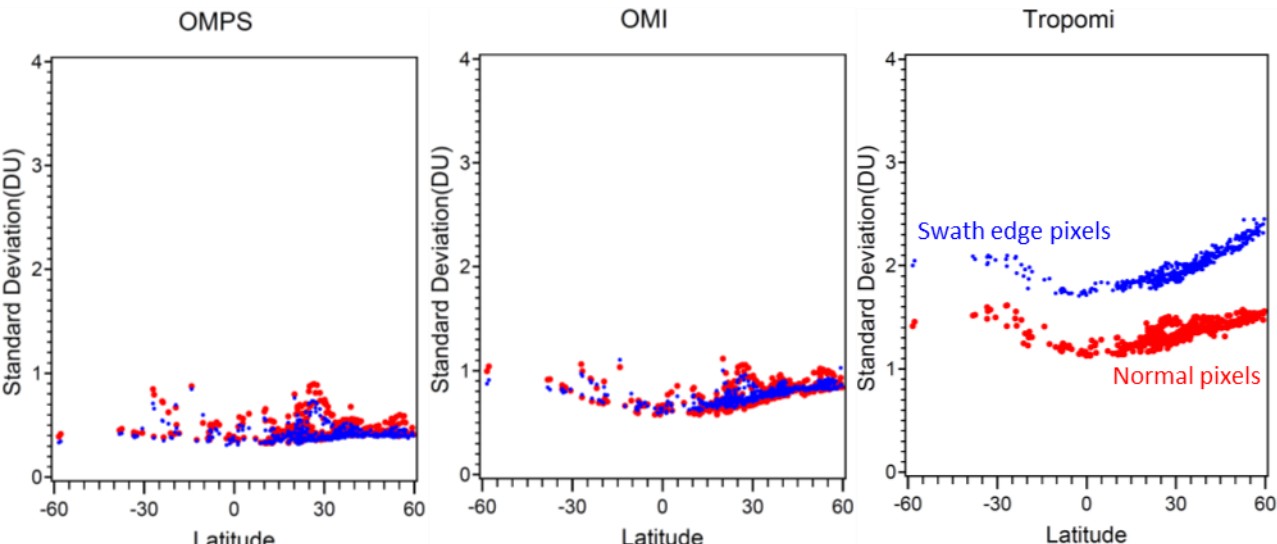

**Figure 2**. The OMPS, OMI, and TROPOMI SO$_2$ standard deviation vs. latitude for "normal" (red) and "swath edge" (blue) pixels. "Swath edge" pixels were defined as the first and the last 3, 5, and 20 pixels for OMPS, OMI, and TROPOMI respectively. All other pixels were considered as "normal". The plot is based on satellite measurements centred between 150 km and 300 km around the sources from the catalogue. The sources under the South Atlantic Anomaly (SAA) are excluded.



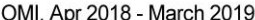

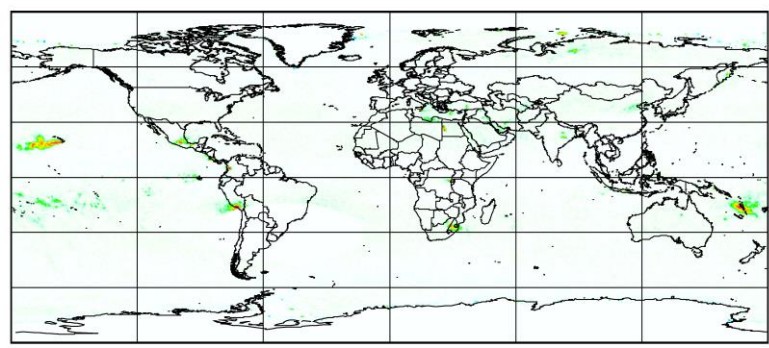

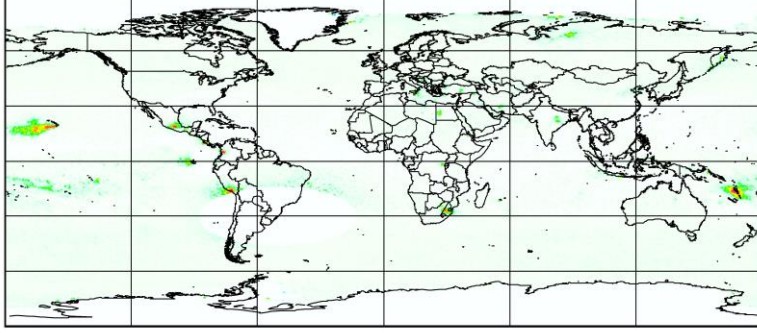

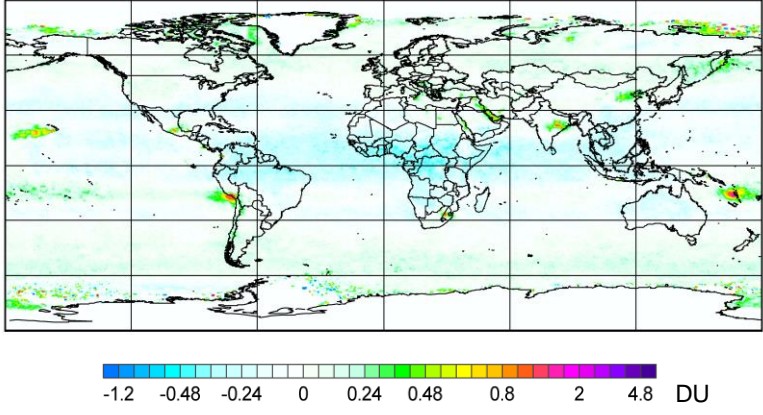

**Figure 3**. Mean SO$_2$ (DU) over the globe from TROPOMI, OMI, and OMPS for the period from April 2018 to March 2019. Data are smoothed by oversampling technique with radius R=30 km. The area of the South Atlantic Anomaly (SAA) is left blank on the OMI and OMPS maps. The SAA largely increases uncertainties of OMI and OMPS SO$_2$ data but has a much smaller effect on TROPOMI SO$_2$ data.

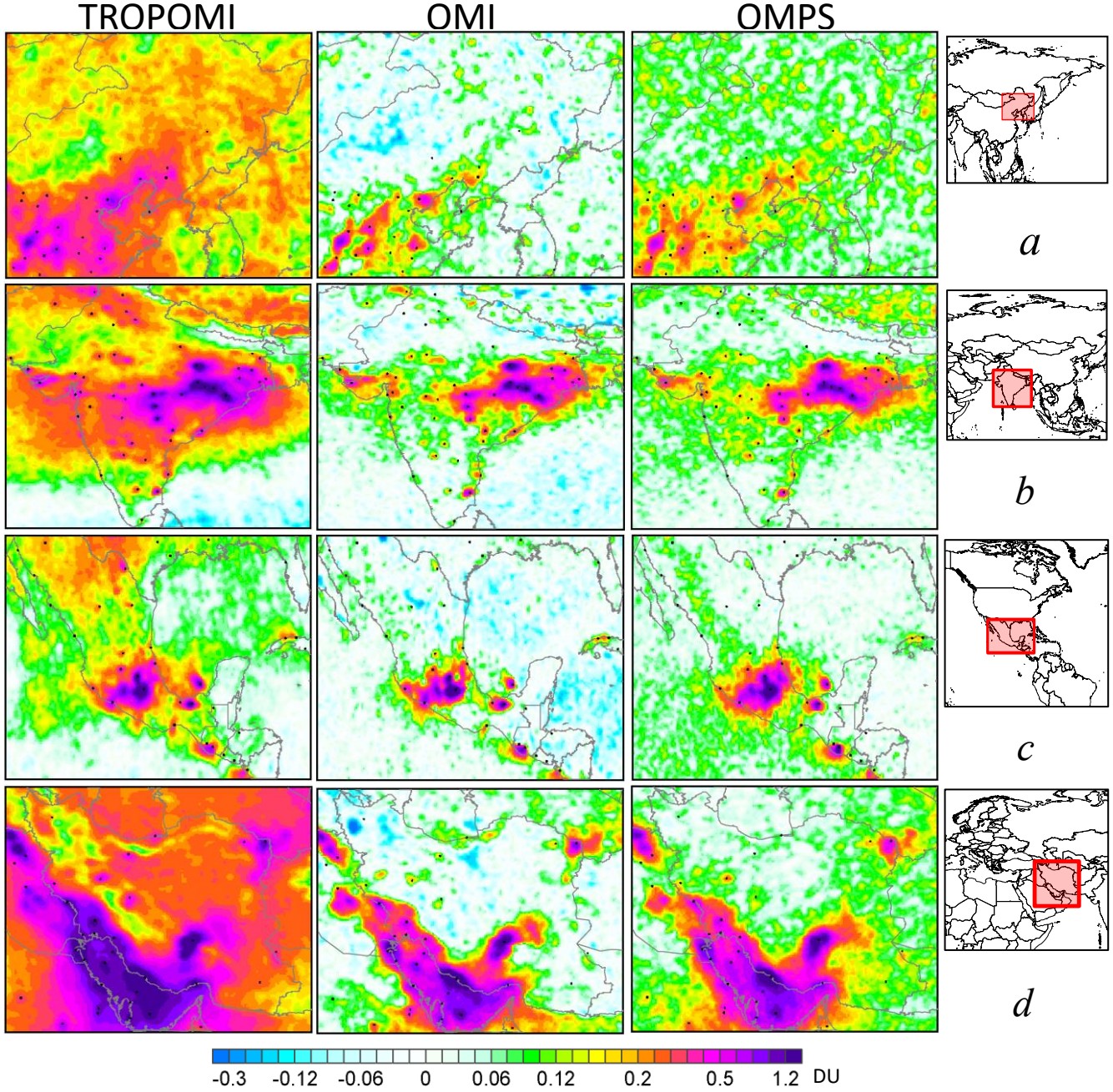

**F**i**gure 4**. Mean $SO_2$ (DU) from TROPOMI, OMI, and OMPS over northern China (a), India (b), Mexico (c), and Iran (d) in the period from April 2018 to March 2019. Large-scale biases make it difficult to interpret Tropomi $SO_2$ data and compare them to OMI and OMPS directly. Data are smoothed by oversampling technique with radius R=30 km. The black dots indicate the $SO_2$ sources. Note that the color scale is different from the scale in Figure 3.





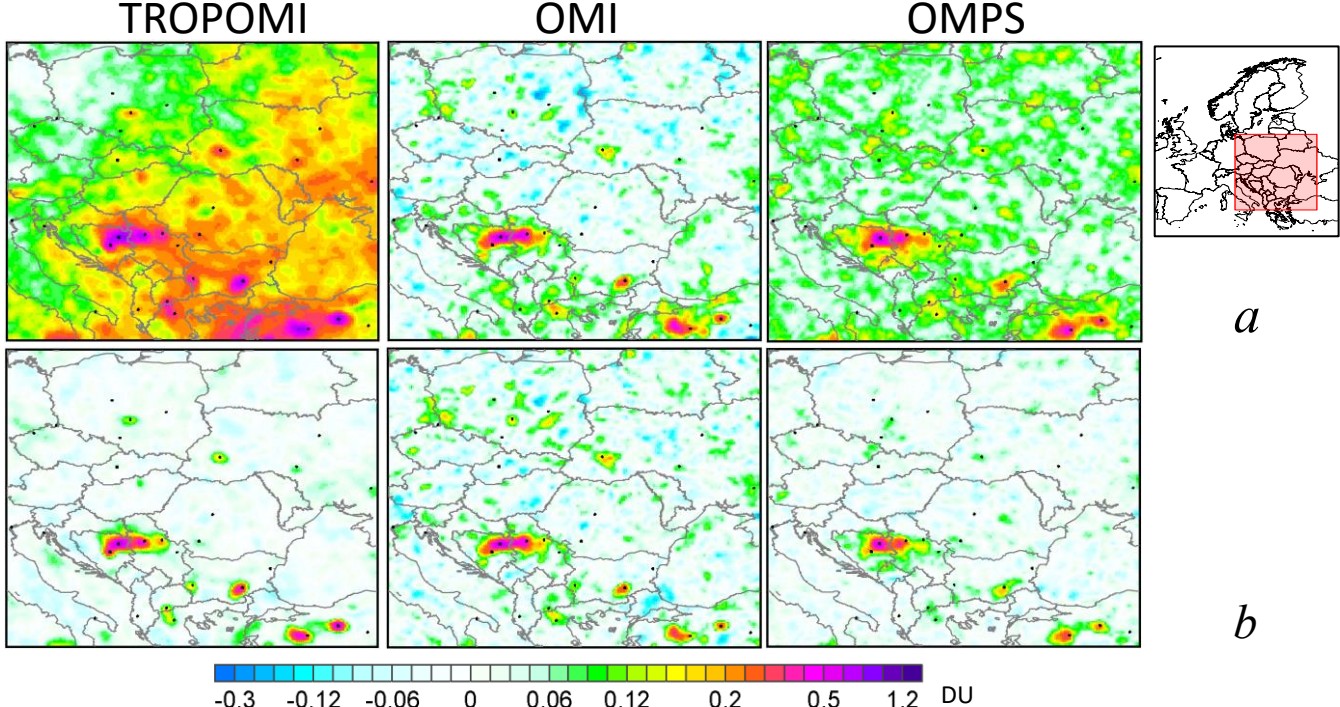

**Figure 5.** TROPOMI, OMI and OMPS mean SO$_2$ over Eastern and Southern Europe in the period from April 2018 to March 2019 (a). The same data, but with large-scale bias removed (b). Data are smoothed by oversampling pixel averaging with R=30 km radius; the black dots indicate the SO$_2$ emission sources.

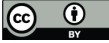

**Figure 6.** SO$_2$ hotspot over Serbia and Bosnia and Hercegovina: mean OMI values over a 5-year period (2014-2018) and mean OMI and TROPOMI values over 1-year period (from April 2018 to March 2019). Data are smoothed by oversampling/pixel averaging with 30 km radius; constant bias is removed; the black dots indicate the emission sources. The top panels show mean SO$_2$, the bottom panel show the ratios of the mean SO$_2$ value to the standard error of the mean.



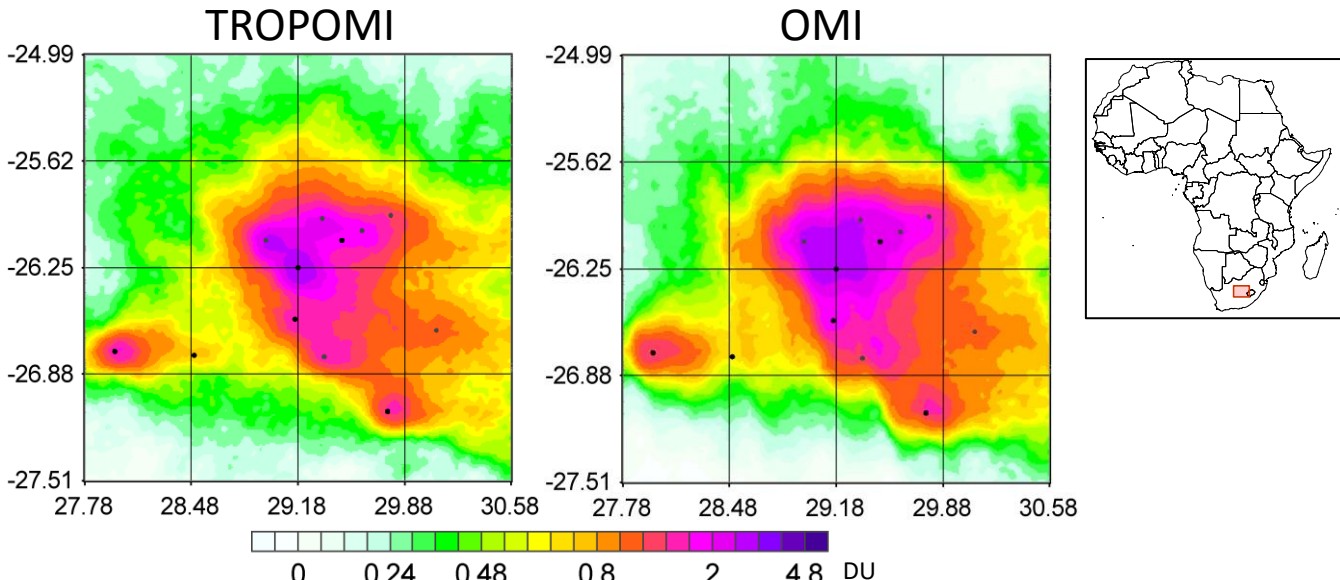

**Figure 7**. The mean TROPOMI SO$_2$ over a cluster of power plants in South Africa in the period from April 2018 to March 2019 and the mean 2005-2019 OMI SO$_2$ over the same region. Data are smoothed by oversampling/pixel averaging with 10 km radius. The black dots indicate the emission sources. Note that the color scale is different from the scale in the previous figures.



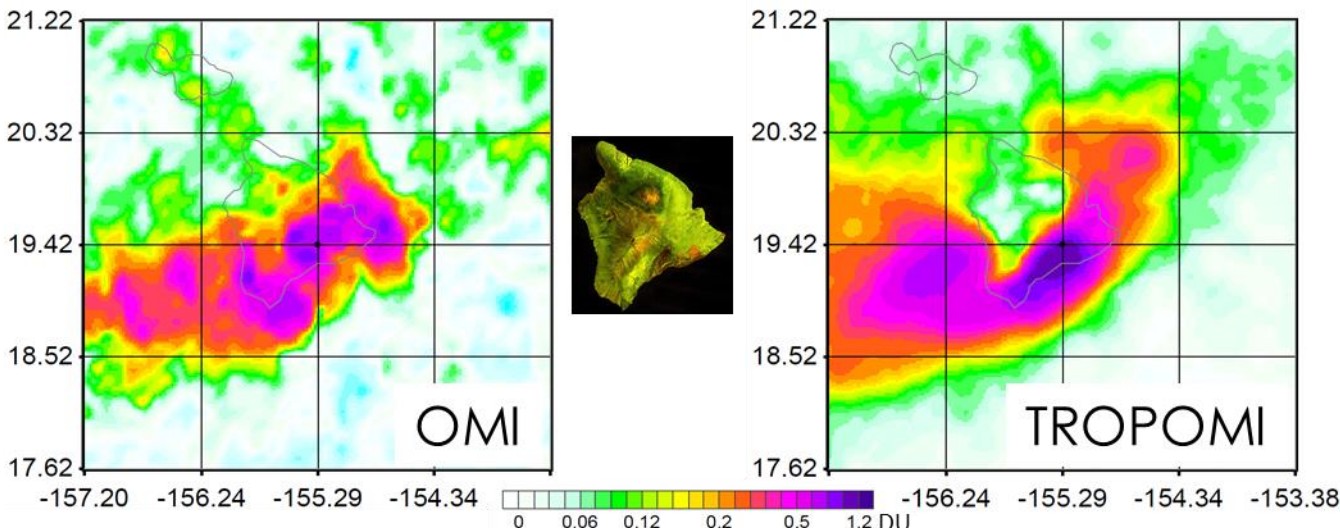

**Figure 8.** Mean SO$_2$ in DU over Kilauea Volcano, Hawaii, from OMI and TROPOMI data for the period from April 2018 to March 2019. The volcano is in the centre of the map. The influence of orography on the SO$_2$ distribution is clearly visible due to the high spatial resolution of Tropomi. A Sentinel-1 image from 23 May 2018 that illustrates the island's orography is shown in the middle of the figure.



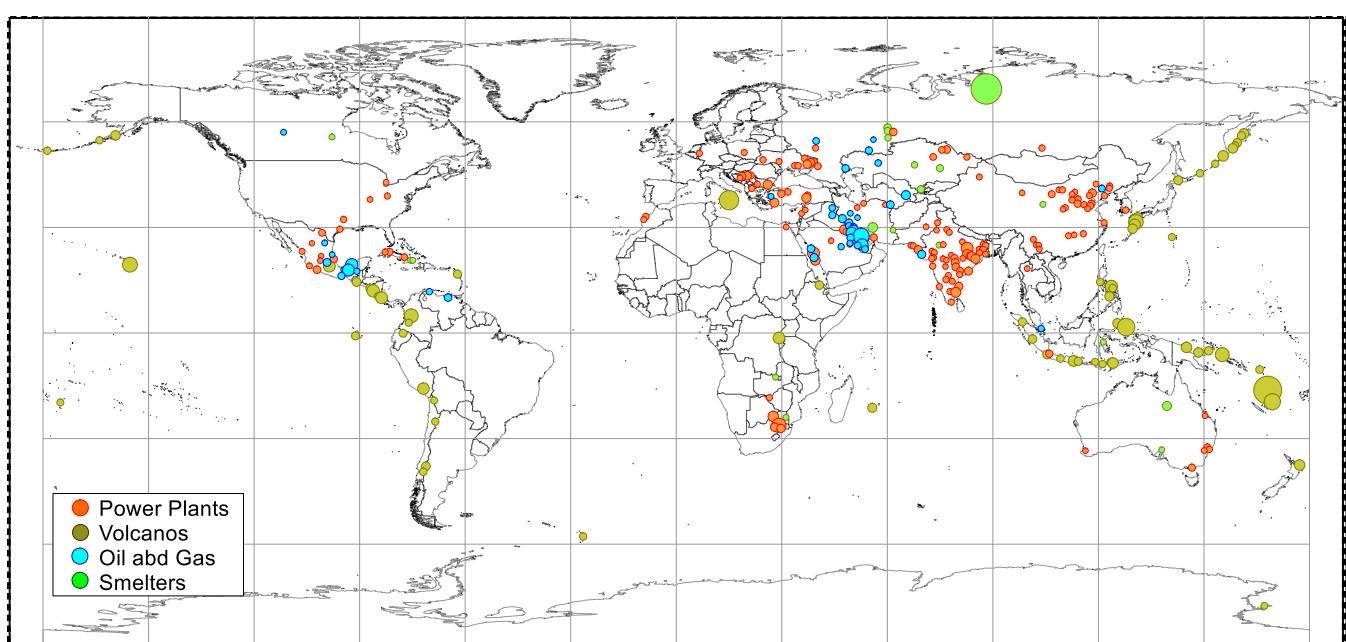

**Figure 9.** SO$_2$ emission sources seen by Tropomi in 2018. We checked ~500 locations where OMI detected SO$_2$ emissions in 2005-2014 (Fioletov et al., 2016). Note that some of them are not active now or have their emissions significantly reduced. Tropomi can "see" 278 sites including 150 Power Plants, 19 Smelters, 41 Oil and Gas industry-relates sources, and 68 Volcanoes with annual emissions from 10 to 2000 kt. The size of the symbols is proportional to the annual emission values.

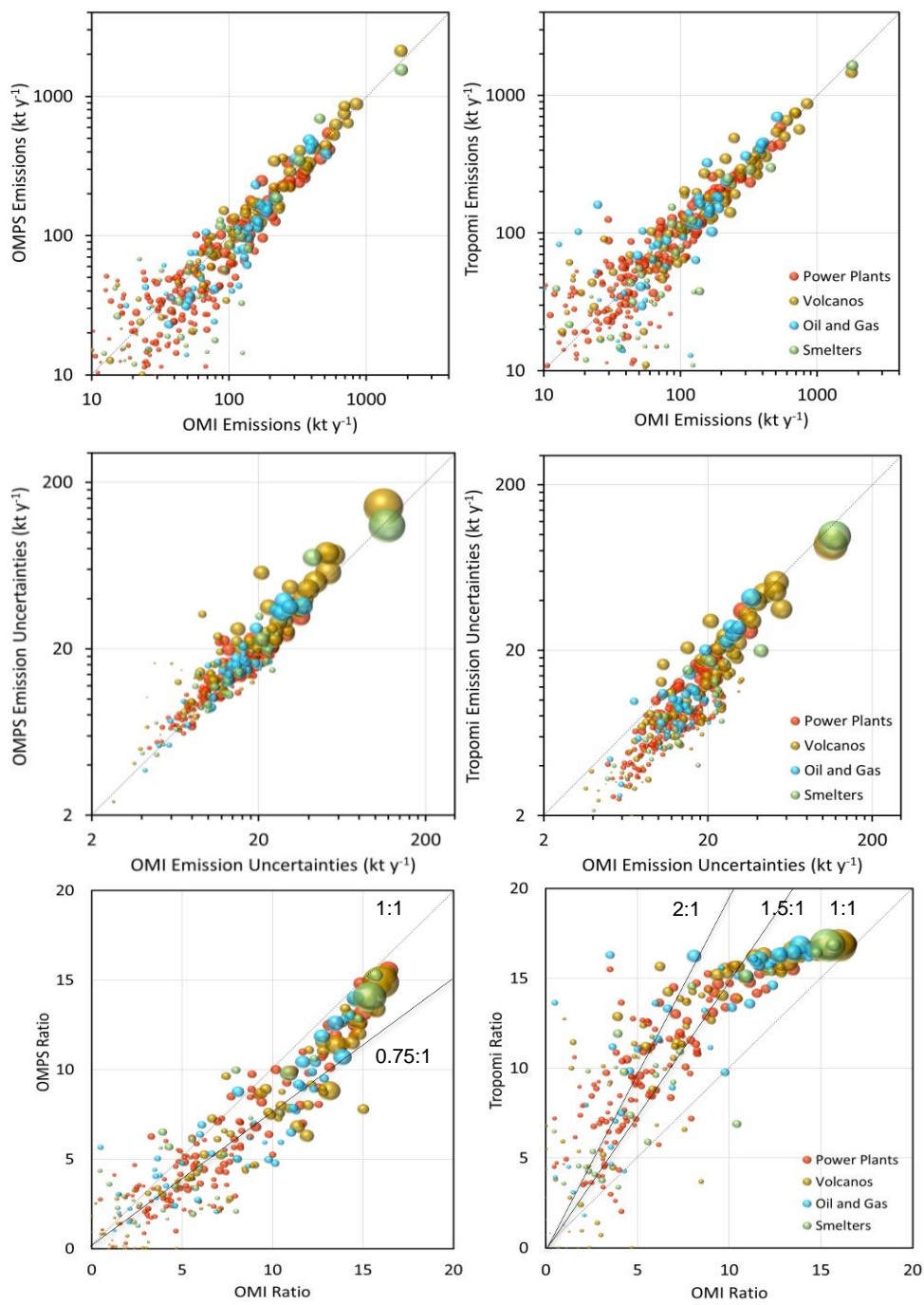

**Figure 10.** (a) OMI, OMPS, and TROPOMI-based estimated emissions in kt per year. The bubble area is proportional to the ratio of emission to uncertainty. The bigger the bubble, the more reliable the estimate. (b) Emission uncertainties in kt per year. The bubble area is proportional to the emission rate. The bigger the bubble the higher emissions. (c) Ratios between estimated emissions and their uncertainties. The bubble area is proportional to the emission rate.



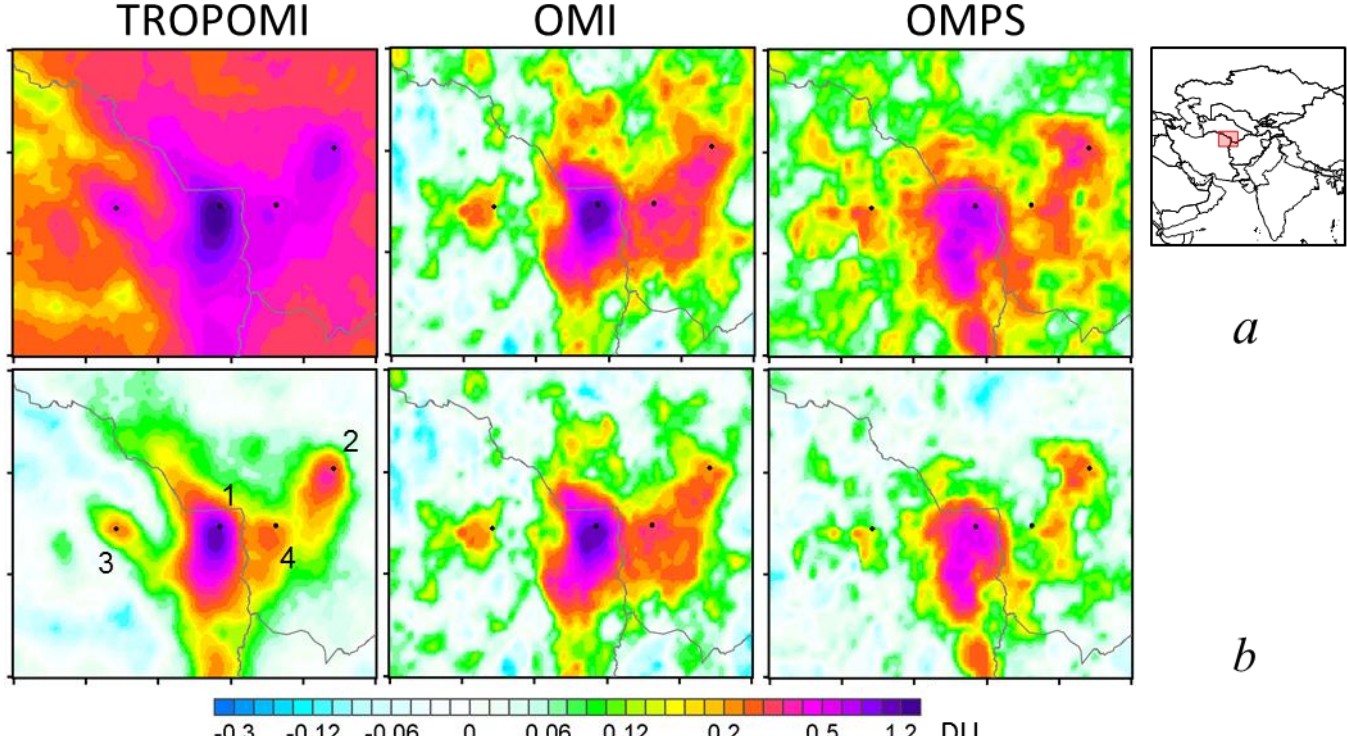

**Figure 11.** TROPOMI, OMI and OMPS mean SO$_2$ (DU) over the Khangiran region for the period from April 2018 to March 2019 (a). The same data, but with large-scale bias removed (b). Data are smoothed by oversampling/pixel averaging with R=20 km radius. The black dots indicate the SO$_2$ emission sources: gas refinery at Khangiran, Iran (1); gas exploration sources at Mary (2) and Sovetabad (4), Turkmenistan; Toos and Ferdosi power plants, Iran (3).

