# Peer review of "Anthropogenic and volcanic point source SO2 emissions derived from TROPOMI onboard Sentinel 5 Precursor: first results"

_Atmospheric Chemistry and Physics, 2019_

## Referee Comment (RC1) · Anonymous Referee #1 · 4 Mar 2020

The authors present in this paper first result for the estimation of SO2 emissions from point sources using TOPOMI data. They apply a well-documented methodology already applied in previous publications on other satellite data (e.g. OMI, OMPS). They demonstrate in this publication the impact of TROPOMI's high spatial resolution on the potential of the algorithm to estimate emissions from weaker sources as well as to finer separation of point sources. Issues relative to large scale biases are discussed and the results are compared with the ones estimated from OMI and OMPS data. The paper is well written and structured and should be accepted for final publication after considering my comments below.

[Figure]

Page 5, Lime 11: The -3DU threshold is related to the spread of the distribution of the SO2 values? Please comment.

Page 5, lines 12-14: Probably the comma is misplaced, but generally this sentence needs rephrasing. It is not clear how this limit is set. Please elaborate here more. As it is now these limits seem arbitrary.

Page 6, Line1-9: The whole discussion on the AMF and the temperature dependence is confusing. Do the authors use a specially processed TROPOMI product for this paper (without temperature adjustment, but increased by 22%) and the official product is still the one where an AMF correction factor for the temperature dependence is applied?

Page 6, line 24-27. Is there any possible explanation for this seasonality in standard deviation?

Page 7, line 5-10. Can the authors provide an explanation, why the standard deviations from TROPOMI are larger? Is it instrumental or a matter of spatial resolution differences?

Page 10, lines 9-16: Is there any justification for the size of the rectangular where the fit is applied and its dependency on the source strength?

---

## Referee Comment (RC2) · Anonymous Referee #2 · 6 Mar 2020

This manuscript provides the first $SO_2$ emissions estimates for ~278 point sources using TROPOMI observations. The method applied has been used before on OMI and OMPS $SO_2$ data. Results are compared with OMI and OMPS estimates, and specific issues related to the higher spatial resolution of TROPOMI and specific biases in TROPOMI are discussed.

The paper is wel suited for this journal. I have the following comments.

- p.5 l. 11-12 'To eliminate ..days then ..' somehow this sentence does not make sense to me.
- P.6 l. 8 what is meant with THIS systematic difference ?
- P.6 l9 is the 22% the result of this different use of absorption cross sections at different temperatures ?
- P.6 l. 23 What do you mean with shorter exposure times at the edge of the swath. This is not correct I think. The SNR could be lower due to less binning across track, but that has nothing to do with shorter exp. times.
- P.8 if the biases are seasonal dependent what is its effect on the emissions calculated using annual means $SO_2$ ?
- Data availability : Should the locations (Fig. 9) and emissions (Fig. 10) not be made available through some database connected to the manuscript ? Otherwise other people can not use those.

---

## Author Comment (AC1) · 1 Apr 2020

The authors present in this paper first result for the estimation of SO2 emissions from point sources using TOPOMI data. They apply a well-documented methodology already applied in previous publications on other satellite data (e.g. OMI, OMPS). They demonstrate in this publication the impact of TROPOMI's high spatial resolution on the potential of the algorithm to estimate emissions from weaker sources as well as to finer separation of point sources. Issues relative to large scale biases are discussed and the results are compared with the ones estimated from OMI and OMPS data. The paper is well written and structured and should be accepted for final publication after considering my comments below.

We would like to thank the reviewer for the evaluation and comments that helped us improve the manuscript.

Page 5, Lime 11: The -3DU threshold is related to the spread of the distribution of the SO2 values? Please comment.

Apparently, large negative values are related to very rare specific conditions (probably due to extremally high HCHO amounts). To determine the threshold, the calculations were done for various cut-off limits from -1 DU to -5 DU. It was found that setting the limit below -3 DU changed the emission values to negative for rare sites where such values <-3DU occur.  Rising the limit above -3DU was changing the emission estimates for most of the sites. Therefore the -3DU value was used. We added a sentence to address this issue:

"Lower than -3 DU threshold values produced negative emission values in some rare cases, while higher values affected the emission estimates themselves."

Page 5, lines 12-14: Probably the comma is misplaced, but generally this sentence needs rephrasing. It is not clear how this limit is set. Please elaborate here more. As it is now these limits seem arbitrary.

To make it clear, we changed the text to:
"To eliminate cases of transient volcanic SO2, days with high volcanic SO2 values were excluded from the analysis. If the highest 10% of SO2 values near the analyzed site were above a certain limit on a particular day, all data from the entire day were excluded. The limit depended on the emission strength and varies from 6 DU for sources emitting less than 100 kt per year to 15 DU for sources emitting >1000 kt per year (see Fioletov et al., (2016) for details)."

Page 6, Line1-9: The whole discussion on the AMF and the temperature dependence is confusing. Do the authors use a specially processed TROPOMI product for this paper (without temperature adjustment, but increased by 22%) and the official product is still the one where an AMF correction factor for the temperature dependence is applied?

To make it clear, we changed the text to:

"In this work, for consistency, we used TROPOMI SO$_2$ SCDs and converted them to VCDs using the same AMF as we utilized for OMI and OMPS (without any temperature adjustment). However, that

meant that the obtained TROPOMI VCDs corresponded to 203 K as the original TROPOMI SCD were calculated for that temperature. To remove the systematic difference with OMI/OMPS data caused by the difference in cross section temperature (203° K for TROPOMI vs. 293° K for OMI/OMPS), we increased the TROPOMI $SO_2$ VCDs by 22% (see Theys et al., (2017), their Figure 6, for justification)."

Page 6, line 24-27. Is there any possible explanation for this seasonality in standard deviation?

It likely has the same explanation as for the difference in the noise between low and high latitudes: low Sun means lower signals particularly due to a longer path through the ozone layer. We change the text to:

"Outside the tropical belt, there is also some seasonality in the standard deviation values with higher values occurring in winter and lower in summer (not shown) due to weaker signals at low Sun."

Page 7, line 5-10. Can the authors provide an explanation, why the standard deviations from TROPOMI are larger? Is it instrumental or a matter of spatial resolution differences?

A smaller size of TROPOMI pixels means that less photons can reach a single cell of the detector compared to OMI and OMIS. As mentioned, if TROPMI pixels are combined to match OMI/OMPS footprints, TROPOMI actually performs better (i.e., measurements have smaller uncertainties) than OMI/OMPS. We added that:

"As Figure 2 shows, the standard deviations ($\sigma$) for TROPOMI are roughly 1.5 time larger than OMI, and 3 times larger than OMPS since the TROPOMI footprint is smaller and each detector cell receives less photons than OMI and PMPS detector cells."

Page 10, lines 9-16: Is there any justification for the size of the rectangular where the fit is applied and its dependency on the source strength?

We added some additional explanation to the text:

"The parameter estimation was done using OMI pixels centered within a rectangular area that spreads ±L km across the wind direction, L km in the upwind direction and 3·L km in the downwind direction. As in the original study, the value of L was chosen to be 30km for small sources (under 100 kt $yr^{-1}$), 50km for medium sources (between 100 and 1000 kt $yr^{-1}$), and 90km for large sources (more than 1000 kt $yr^{-1}$). For small sources, different L values have little effect on the estimated parameters, but smaller values of L allow to separate individual sources where multiple sources are located in the same area. For larger sources, pixels with elevated SO2 values are located over larger areas and therefore the parameters estimated for higher L values have smaller uncertainties."

---

## Author Comment (AC2) · 1 Apr 2020

This manuscript provides the first $SO_2$ emissions estimates for ~278 point sources using TROPOMI observations. The method applied has been used before on OMI and OMPS $SO_2$ data. Results are compared with OMI and OMPS estimates, and specific issues related to the higher spatial resolution of TROPOMI and specific biases in TROPOMI are discussed. The paper is wel suited for this journal. I have the following comments.

We would like to thank the reviewer for the evaluation and comments that helped us improve the manuscript.

- p.5 l. 11-12 'To eliminate ..days then ..' somehow this sentence does not make sense to me.

It was a typo, it should be "when" instead of "then". Corrected.

- P.6 l. 8 what is meant with THIS systematic difference ?

We changed the text to:

"To remove the systematic difference with OMI/OMPS data caused by the difference in cross section temperature (203 K for TROPOMI vs. 293 K for OMI/OMPS), we increased the TROPOMI $SO_2$ VCDs by 22% (see Theys et al., (2017), their Figure 6, for justification). "

- P.6 l9 is the 22% the result of this different use of absorption cross sections at different temperatures ?

Yes. We edited that paragraph to make the cross-section temperature difference clear.

- P.6 l. 23 What do you mean with shorter exposure times at the edge of the swath. This is not correct I think. The SNR could be lower due to less binning across track, but that has nothing to do with shorter exp. times.

We agree with the reviewer and changed the text as suggested.

- P.8 if the biases are seasonal dependent what is its effect on the emissions calculated using annual means $SO_2$ ?

The actual calculations were done for individual seasons. We added the explanation to page 10:

"To remove the local biases mentioned above, the average $SO_2$ VCD for the area located upwind from the source was calculated and then subtracted from the data. As the biases may be different from season to season, all calculation were done for 3-month periods (seasons) and then the annual emission rate was calculated by averaging seasonal emission rates. "

- Data availability: Should the locations (Fig. 9) and emissions (Fig. 10) not be made available through some database connected to the manuscript ? Otherwise other people can not use those.

We added a supplement with this information. Note that we also re-examined the set of emission sources and excluded four additional sources with the ratio of the estimated emission to its uncertainty below 4.